## [**Supplementary Information**: Record of Transparent Peer Review · Advanced Genetics]

Advancing the genetics of Lewy body disorders with disease-modifying treatments in mind

Gilberto Levy*, Bruce Levin, Eliaz Engelhardt

* Corresponding author

Review timeline:	Date submitted:	09-Feb-2022
	1 st Editorial Decision:	01-Jun-2022
	Revision Received:	22-Jun-2022
	2 nd Editorial Decision:	12-Jul-2022
	Revision Received:	13-Jul-2022
	Accepted:	14-Jul-2022

Editor: Myles Axton/Kerstin Brachhold

1st Peer Review

09-Feb-2022 to 01-Jun-2022

Reviewer #1

The manuscript by Levy et al entitled "A nosological argument for advancing the genetics of Lewy body disorders" is a perspective on how to categorize Lewy body disorders in the post-genome era.

The authors present a component cause model and evolution-based model of causation as a toolbox or framework to define these diseases.

While the authors lay out the challenges in the field and provide a narrative about controversies, the article rather adds confusion than clarifying and finding solutions, due to the writing style. The text needs more explanations and definitions instead of assuming the reader knows all details about classifications and consortium guidelines, since the manuscript is prepared for a genetics journal and not a specialized movement disorder journal.

The manuscript needs careful editing and re-write with more focus on details and the proposed models are accepted, but putting it to practice for PD/PDD/DLB/AD/other neurodegenerative disorders is the key contribution of an article like this.

1. define diseases that need clarification - The article starts with DLB/PDD and ends with 'neurodegenerative diseases'. While the title suggests the focus on Lewy body disorders, in the text Alzheimer/taupathies and 'other neurodegenerative disorders' are discussed. What is the message here?
2. Need a table or box with clinical and neuropathological definition of diseases discussed in article.
3. With the more recent clinical acceptance of dementia in PD, there has been more openness to accept genetic markers of Alzheimer for Lewy body disorders and studies combining candidate genes for both alpha-synucleinopathies and taupathies. Discuss as gene candidate studies include both "AD" and "PD" risk factors and its rationale.

4. While 20 Mendelian forms of 'PD' are discussed (and yes they fit the 'one' cause), however, a closer look at the monogenic causes of PD indicates that those 'Mendelian forms of PD' are different diseases with 'parkinsonism' as a clinical feature, e.g. as described in Langston et al. 2015, Figure 1. This fact needs to be critically discussed instead of just taken for granted that a PARK loci means Parkinson's disease, e.g. PARK9 is Kufor-Rakeb syndrome (autosomal recessive young onset Parkinson's disease (YOPD), spastic paraparesis, abnormal eye movements and facial myokymia), very far from PD or Lewy body disease.
5. The manuscript would benefit from subheaders
6. Consider terms - lumping/splitting for combine/separate
7. p.6: SCC and EBMC model should provide focused examples for DLB/PDD genes/symptoms/neuropathology
8. Define key genetic terms: SNP, variant, mutation. Which genetic term is used to describe a pathogenic genetic change and contributes to disease?
9. p.10/11: This is the key section for the argument that DLB and PD have a different genetic profile. Please add a table with the different studies discussed including columns for sample size, ethnicity, clinical or neuropath diagnosis, SNPs and candidate genes tested, similar to Chittor-Vinod et al. 2021, Table 3, Int. J. Mol. Sci. 2021, 22, 1045. <https://doi.org/10.3390/ijms22031045>
10. Title should be revised to fit the content of article.
- 11.p.4 intro is interesting, could be strengthened by adding a timeline of this evolution/definitions, when where these consortium meetings.
12. "We proposed elsewhere a data-driven alternative to the one-year rule.[26]"
What is this? Context, would be good to describe briefly the alternative to keep the reader focused on the manuscript.
13. Consider meaningful term: not "The nosological argument"

Reviewer #2

Levy and colleagues offered an interesting perspective on a long-debated topic, if to consider Lewy body dementia and Parkinson's disease as different entities.

the semi-structured manuscript offers an good overview on pros and cons, delving into details on the genetic findings and less on other aspects (neuropathology, clinical presentations etc) that I suggest to expand.

I would also improve the conclusions, where the perspective of the authors appear weak and ultimately doesn't offer a strong take on the matter.

1 st Editorial Decision	01-Jun-2022
Editorial Decision: Revise and resubmit after addressing the reviewers' comments	
Recommendation of the reviewers	
Reviewer #1 Recommends Major Revision	
Reviewer #2 Recommends Major Revision	

Reviewer #1

“1. define diseases that need clarification - The article starts with DLB/PDD and ends with 'neurodegenerative diseases'. While the title suggests the focus on Lewy body disorders, in the text Alzheimer/taupoathies and 'other neurodegenerative disorders' are discussed. What is the message here?”

The focus of the article is indeed on Lewy body disorders. We now removed any mention to AD or another neurodegenerative disease that is not essential. In reviewing the literature, some mentions are unavoidable, either because several studies included data on these other diseases (in addition to DLB and PD) or because the relation between DLB and AD serve as a reference against which to evaluate the relation between DLB and PD.

“2. Need a table or box with clinical and neuropathological definition of diseases discussed in article.”

Now that the focus of the article on Lewy body disorders has been sharpened, and the distinction between DLB and PDD is described in the text, we believe that this is unnecessary.

“3. With the more recent clinical acceptance of dementia in PD, there has been more openness to accept genetic markers of Alzheimer for Lewy body disorders and studies combining candidate genes for both alpha-synucleinopathies and taupoathies. Discuss as gene candidate studies include both 'AD' and 'PD' risk factors and its rationale.”

Consistent with the focus of the article on Lewy body disorders, we selected and discussed studies with information relevant to the possibility of widespread sharing of genetic factors between DLB and PD. We now found one additional study that was not reviewed in the previous version of the manuscript, by Bras et al. (2014), and added it to the discussion.

“4. While 20 Mendelian forms of 'PD' are discussed (and yes they fit the 'one' cause), however, a closer look at the monogenic causes of PD indicates that those 'Mendelian forms of PD' are different diseases with 'parkinsonism' as a clinical feature, e.g. as described in Langston et al. 2015, Figure 1. This fact needs to be critically discussed instead of just taken for granted that a PARK loci means Parkinson's disease, e.g. PARK9 is Kufor-Rakeb syndrome (autosomal recessive young onset Parkinson's disease (YOPD), spastic paraparesis, abnormal eye movements and facial myokymia), very far from PD or Lewy body disease.”

As highlighted in the revised version of the manuscript (page 7), we now noted that “Mendelian forms of PD can have characteristic manifestations in addition to parkinsonism”. We believe that an extensive discussion of the clinical characteristics of Mendelian forms of PD is beside the point. It does not have an impact on and would represent a distraction from the central argument of the manuscript, pertaining to

the elucidation of genetic causation in the context of a nosological controversy.

“5. The manuscript would benefit from subheaders”

We thank the reviewer for this suggestion. We now added the headers and subheaders 2, 3, 3.1, and 3.2.

“6. Consider terms - lumping/splitting for combine/separate”

Lumping/splitting are standard terms in the nosology literature.

“7. p.6: SCC and EBMC model should provide focused examples for DLB/PDD genes/symptoms/neuropathology”

This is done later in the text, as highlighted on page 8, and in the legend of Figure 1.

“8. Define key genetic terms: SNP, variant, mutation. Which genetic term is used to describe a pathogenic genetic change and contributes to disease?”

We appreciate the reviewer’s intent to make the manuscript more accessible, but believe this is unnecessary for the audience of the journal.

“9. p.10/11: This is the key section for the argument that DLB and PD have a different genetic profile. Please add a table with the different studies discussed including columns for sample size, ethnicity, clinical or neuropath diagnosis, SNPs and candidate genes tested, similar to Chittor-Vinod et al. 2021, Table 3, Int. J. Mol. Sci. 2021, 22, 1045. <https://doi.org/10.3390/ijms22031045>”

We thank the reviewer for this suggestion. We now added Table 2 to the manuscript, which summarizes genetic studies on DLB relevant to the investigation of genetic sharing between DLB and PD.

“10. Title should be revised to fit the content of article.”

We revised the title as requested. Now the title is “Advancing the genetics of Lewy body disorders with disease-modifying treatments in mind”. This new title makes it explicit that the central argument of the manuscript, pertaining to the elucidation of genetic causation in the context of a nosological controversy, takes into account the ultimate aim of developing disease-modifying treatments.

“11.p.4 intro is interesting, could be strengthened by adding a timeline of this evolution/definitions, when where these consortium meetings.”

We agree with this point and now added throughout the text the years when the consensus reports were published.

“12. ‘We proposed elsewhere a data-driven alternative to the one-year rule.[26]’

What is this? Context, would be good to describe briefly the alternative to keep the reader focused on the manuscript.”

We did that, as highlighted on page 4.

“13. Consider meaningful term: not ‘The nosological argument’”

We are now using “nosological controversy”.

Reviewer #2

“Levy and colleagues offered an interesting perspective on a long-debated topic, if to consider Lewy body dementia and Parkinson's disease as different entities. the semi-structured manuscript offers an good overview on pros and cons, delving into details on the genetic findings and less on other aspects (neuropathology, clinical presentations etc) that I suggest to expand.”

We thank the reviewer for recognizing the strengths of the manuscript. The central argument of the manuscript pertains to the elucidation of genetic causation, hence the focus on genetic findings. There are many good reviews on the neuropathological and clinical similarities and differences between DLB and PD, which we quote in the text and in Table 1.

“I would also improve the conclusions, where the perspective of the authors appear weak and ultimately doesn't offer a strong take on the matter.”

As requested by the reviewer, we rewrote the conclusion in order to make it stronger (highlighted in the manuscript). We now further emphasize the implications of the central argument of the manuscript for the ultimate goal of developing disease-modifying treatments for Lewy body disorders. We also rewrote the abstract to reflect that.

2 nd Peer Review	22-Jun-2022 to 12-Jul-2022
----------------------------

Reviewer #1

The response of the authors only partially responded to the critiques. The newly added Table 2 shows the study design, but not the summary of results (add another column). Otherwise, ok to publish.

Second Editorial Decision	12-Jul-2022
Editorial Decision: Revise and resubmit after addressing the final minor comments of reviewer #1	
Recommendation of the reviewers	
Reviewer #1 Recommends Minor Revision	

Authors' Response to 2nd Review

13-Jul-2022

We included one additional column in Table 1, with the main findings from the studies, as requested by the reviewer.

Final Decision

14-Jul-2022

Accept the revised version for publication as the authors satisfactorily addressed the final minor comments of reviewer #1.